# Efficacy of Whole-Blood Del Nido Cardioplegia Compared with Diluted Del Nido Cardioplegia in Coronary Artery Bypass Grafting: A Retrospective Monocentric Analysis of Pakistan

**DOI:** 10.3390/medicina57090918

**Published:** 2021-08-31

**Authors:** Adnan Haider, Irfan Azmatullah Khwaja, Ammar Hameed Khan, Muhammad Shahbaz Yousaf, Hafsa Zaneb, Abdul Basit Qureshi, Habib Rehman

**Affiliations:** 1Department of Physiology, University of Veterinary and Animal Sciences, Lahore 54000, Pakistan; adnanhaider.ccp@gmail.com (A.H.); drmshahbaz@uvas.edu.pk (M.S.Y.); 2Department of Cardiovascular Surgery, King Edward Medical University, Lahore 54000, Pakistan; Irfan_a_khwaja@hotmail.com; 3Department of Cardiovascular Surgery, Shalamar Medical and Dental College, Lahore 54812, Pakistan; ammarhameed@hotmail.com; 4Department of Anatomy and Histology, University of Veterinary and Animal Sciences, Lahore 54000, Pakistan; hafsa.zaneb@uvas.edu.pk; 5Department of Surgery, Services Institute of Medical Sciences, Lahore 54810, Pakistan; basitq@hotmail.com

**Keywords:** cardiac surgery, myocardial protection, aortic cross-clamp time, Troponin-T

## Abstract

*Background and Objectives*: Cardioplegia is one of the most significant components used to protect the myocardium during cardiac surgery. There is a paucity of evidence regarding the utilization of whole-blood Del Nido cardioplegia (WB-DNC) on clinical outcomes in coronary artery bypass grafting (CABG). The purpose of this retrospective cross-sectional study is to compare the effectiveness of diluted (blood to crystalloid; 1:4) Del Nido cardioplegia (DNC) with WB-DNC in patients who underwent elective CABG in a tertiary care hospital in Lahore-Pakistan. *Materials and Methods*: This was a retrospective descriptive study conducted at the Department of Cardiovascular Surgery, King Edward Medical University, Lahore. The medical database of all consecutive patients admitted from January 2018 to March 2020 and who fulfilled the inclusion criteria were reviewed. *Results*: Out of 471 patients admitted during the study period, 450 underwent various elective cardiac surgeries. Out of 450, 321 patients (71.33%) were operated on for CABG. Only 234/321 (72.89%) CABG patients fulfilled our inclusion criteria; 120 (51.28%) patients received WB-DNC, while 114 (48.71%) patients were administered with DNC. The former group presented with better clinical outcomes compared with the latter in terms of lesser requirements of inotropic support, low degree of hemodilution, shorter in-hospital stay, improved renal function, and cost-effectiveness. Peak values of serum Troponin-T (Trop-T), creatine kinase-myocardial band (CK-MB) release, and activated clotting time (ACT) were also lower in the WB-DNC group compared with the DNC group. *Conclusions*: The WB-DNC conferred better myocardial protection, improved early clinical outcomes, and also proved to be economical for patients undergoing elective CABG compared with classical crystalloid cardioplegia solution.

## 1. Introduction

Recovery from ischemia and functional sustainability of the heart in patients following cardiac surgery using cardiopulmonary bypass (CPB) greatly depends upon better myocardial protection [1]. Cardioplegia is commonly used to induce and provide a depolarized diastolic cardiac arrest [2]. In the early 1990s, Del Nido cardioplegia (DNC) formulated for neonatal and pediatric cardiac surgery was introduced [3] which consists of a calcium-free and potassium-rich non-glucose-based solution with electrolyte composition identical to that of extracellular fluid, and provides longer duration of cardiac arrest [4]. The ingredients of DNC are mixed with highly oxygenated blood and crystalloid solution with a ratio of 4 times crystalloid to 1 times blood [5]. However, the Brigham group used it with a ratio of 8:1 (blood to crystalloid) composition [6]. Due to the ease of administering, single induction dose, and the value of decreasing surgical interruption, its use in adult cardiac surgery gained acceptance [4]. However, crystalloid part of DNC may be linked with hemodilution that causes problems in blood conservation strategy. This disadvantage can be resolved by substituting the crystalloid part of DNC with whole blood being drawn from the oxygenator [7]. Whole-blood DNC has many additional attributes such as free radical scavenging, better oxygen delivery, and reduction in the degree of hemodilution and myocardial edema [8]. Various studies showed that whole-blood cardioplegia has added advantages over traditional cardioplegia in terms of shorter ICU stay and in-hospital stay, infrequent incidences of postoperative low cardiac output syndrome, low postoperative renal injury, and better myocardial protection in CABG surgery [9,10,11]. On the other hand, some reports described higher rate of acute renal injury, in-hospital mortality, and blood transfusion in elderly patients that received whole-blood cardioplegia compared with diluted cardioplegia [12].

As mentioned earlier, several studies have evaluated the use of DNC or WB-DNC separately in cardiac surgery. To the best of our knowledge, there is no study that compared the use of these two Del Nido-based cardioplegia solutions for adult CABG patients in the same hospital setting. Therefore, this retrospective cross-sectional observational study was designed to compare the efficacy and safety of the DNC with WB-DNC in adult patients operated for elective CABG procedures. The medical records were compared for primary (degree of hemodilution, inotropic support, in-hospital stay, cardiac arrhythmias), and secondary outcomes (peak Troponin-T (Trop-T, creatine kinase-myocardial band (CK-MB) levels, aortic cross-clamp (ACC) time, cardiac pulmonary bypass time, serum creatinine).

## 2. Materials and Methods

### 2.1. Study Design and Patients

The study was a cross-sectional observational study and we retrospectively reviewed the database of all patients brought to the Department of Cardiovascular Surgery, King Edward Medical University, Lahore-Pakistan, a tertiary care hospital, for various cardiac surgical interventions from January 2016 to 13 March 2020. From 13 March 2020, the elective cardiac surgeries were temporarily stopped due to the pandemic outbreak of COVID-19. The approval for the study was received from the Ethical Committee of the Department of Cardiovascular Surgery, King Edward Medical University/Mayo Hospital, Lahore-Pakistan (Number: 143/CS/KEMU/2018, dated 26 February 2018). Based on the medical records, patients having documented history of severe pulmonary hypertension, preoperative uncontrolled diabetes, chronic renal failure requiring hemodialysis, poor left ventricular function (ejection fraction <30%), preoperatively on intra-aortic balloon pump, and brought for emergency procedures were excluded from the study. The choice of cardioplegia used was based only on the surgeon’s decision. Out of a total of 471 patient records evaluated during the study period, 450 (95.94%) underwent elective cardiac surgery. Within this cohort, 321 (71.33%) patients were surgically treated for CABG. Out of which, 234 (72.89%) patients fulfilled our inclusion criteria. The number of patients grafted with a single vessel, two vessels, three vessels, four vessels, and five vessels were 14, 95, 94, 27, and 4 respectively. The medical records showed that 120 (51.28%) patients received DNC, while 114 (48.71%) patients were administered with WB-DNC (Figure 1).

### 2.2. Preoperative Details

Pre-operative demographic data including, name, gender, age, weight, height, body mass index, history of smoking, and presence of diabetes mellitus were retrieved from the database. The Society of Thoracic Surgeons National Cardiac Surgery Database definitions were used for this study [6]. Prolonged ventilation was defined as “pulmonary insufficiency requiring a ventilator for >24 h, postoperatively”. EuroSCORE-II was used to predict in-hospital mortality after cardiac surgery in the present study [13]. Baseline parameters including hemoglobin, platelet count, white blood cell count, serum glutamic pyruvic transaminase (SGPT), serum glutamic oxaloacetic transaminase (SGOT), alkaline phosphatase, serum creatinine, blood urea, and left ventricular ejection fraction were extracted from the medical records.

### 2.3. Perioperative Details

Medical records showed that all cardiac surgeries were performed through the total median sternotomy using extracorporeal circulation and moderate hypothermia (26–32 °C) upon the surgeon’s preference with the same perfusion protocols. Briefly, general anesthesia was induced and maintained with midazolam, propofol, and inhaled isoflurane. The CPB circuit was primed with 1.2 to 1.8 L of Ringer lactate solution having 0.25 g/kg mannitol and 3.0 IU/mL of porcine heparin. All CPB were performed using a roller pump (Maquet, HL-20, Harlingen, Germany) and a hollow fiber adult membrane oxygenator (Medtronic Inc., Minneapolis, MN, USA) with a perfusion flow rate of 2.4 L/min/m^2^ of body surface area. Under the direction of the surgeon, a serum hemoglobin level of ≤6.0 g/dL or a hematocrit of ≤18% was considered as an indicator of blood transfusion. The CPB time, aortic cross-clamp (ACC) time, degree of hemodilution in terms of hemoglobin concentration, and urine output were noted. Anticoagulation status in terms of activated clotting time (ACT) was first measured after 30 min of CPB initiation and then after protamine delivery.

### 2.4. Cardioplegia Protocol

In our Institute, the WB-DNC are generally prepared with whole blood by mixing minimal quantities of arresting and additive agents. The ingredients of WB-DNC were constituted in two different 50 mL syringes. First syringe contained 6.6 mL magnesium sulfate (50%), 21.4 mL mannitol (25%), 20.6 mL NaHCO_3_ (8.4%), and 11.4 mL lidocaine (2%). The other 50 mL syringe contained only 50 mEq KCl. Delivery of filled ingredients was carried out through infusion pumps at a rate of 380 mL/hour at 6 °C (Table 1). After applying ACC, an amount of 1.0 L of cardioplegia was delivered as a single dose through the aortic root via antegrade cardioplegia cannula within 3–4 min at a line pressure of 250–300 mm Hg along with a pump flow rate of 300 mL/min and aortic root pressure of 50–60 mm Hg. All distal anastomoses were performed with the single cross-clamp technique. Cold DNC (one part of the blood being drawn from the oxygenator and four parts of Ringer lactate) was administered with the delivery system having the same temperature, flows, and pressure conditions as described above for WB-DNC delivery. It is a normal practice to administer a ‘hot shot’ of 400–500 mL of warm blood of the respective patient before removal of the aortic cross-clamp.

### 2.5. Post-Operative Details

After shifting to ICU, the post-operative variables including peak serum concentrations of cardiac enzymes Trop-T and CK-MB were measured using commercial kits as per the manufacturer’s recommendation (DiaSys Diagnostic Systems GmbH, Holzhein, Germany). Similarly, SGPT and SGOT, alkaline phosphatase, serum creatinine, and blood urea were also measured using commercial kits (Human GmbH, Wiesbaden, Germany). Time to extubation, total inotropic support, length of ICU stay, in-hospital stay, chest tube drainage (blood loss), the number of patients receiving a blood transfusion, ACT, leucocyte, and lowest platelets counts were retrieved from the medical records during the ICUstay. Mortality was defined as death within 30 days of the surgical procedure [14]. Day 1 post-operative data regarding the blood variables were collected from the database.

### 2.6. Study Endpoints

The primary endpoints of the study were degree of hemodilution (hemoglobin values before, during, and after CPB), in-hospital stay, total inotropic support and rhythm disturbances (spontaneous recovery of heart rhythm, electrical cardioversion needed, and pacing).

The secondary endpoints included were surrogates of myocardial protection comprising peak values of cardiac enzymes (serum Trop-T and CK-MB) release, CPB time, ACC time, and renal injury (peak serum creatinine value).

However, as per hospital requirements, other parameters noted were extubation time, blood loss, ACT values, mortality, cost of cardioplegia solution, chest tube drainage, and ICU stay.

### 2.7. Statistical Analysis

Data were analyzed using the SPSS version-26 (SPSS, Inc., Armonk, NY, USA). The normality of the data was determined using the Kolmogorov–Smirnov test. Quantitative variables were presented as means ± SD, while median and range were used to express the data that were not normally distributed. Qualitative variables such as gender, smoking, and diabetes were presented as frequency and percentage. Means/medians were compared using independent Student’s t-test or Mann–Whitney U-test, whichever was applicable. Categorical variables between the two groups were compared using Pearson’s Chi-square test. A probability value of less than 0.05 was considered significant. *p*-value was adjusted for multiple primary and secondary endpoints and was set at *p* = 0.0125 as already described [15].

## 3. Results

Pre-operative patient characteristics and between-group comparisons are presented in Table 2. All the attributes were similar in both the groups except blood urea which was higher (*p* < 0.01) in the WB-DNC group. Post-operative patient characteristics and comparisons between both groups are shown in Table 3.

For primary endpoints, the degree of hemodilution in terms of hemoglobin concentrations peri-operatively, soon after weaning off CPB and shifting to the ICU, were improved (*p* = 0.001) in the WB-DNC group than in the DNC group (Table 3 and Figure 2). The in-hospital stay was prolonged (*p* = 0.001) in the patients who received DNC than the patients maintained with the WB-DNC (Table 3). The inotropic support in terms of adrenalin infusion rate was less (*p* < 0.001) in the WB-DNC group than in the DNC group while in terms of dopamine infusion rate, there was not a statistically significant difference in both groups (Table 3). In our cohort, there was a high percentage (>90%) of spontaneous recovery and of heart rhythm after the release of the ACC without any significant difference between the groups (Table 3). We did not observe any significant differences between the groups regarding atrial defibrillation, pacing, amount of blood loss, blood transfusion, ventilation support, mortality and, ICU stay (Table 3). Peri-operatively, ACT after 30 minutes of initiation of CPB was lower (*p* < 0.01) in the WB-DNC group than in the DNC group (Figure 2), but this effect could not be observed after the surgery (Table 3).

As far as secondary endpoints are concerned, the ACC time and CPB time remained similar in all the patients (Figure 2). Similarly, number of the patients having ACC time with >60 min (74 vs. 66; *p* = 0.600) or longer CPB time (>120 min) were similar in both groups (35 vs. 30; *p* = 0.367). During the perioperative phase, urine output was lower (*p* < 0.001) in the WB-DNC group compared with the DNC group (Figure 2). The average cost for one liter of the DNC was USD 5.50 compared to WB-DNC which was USD 5.00 (since the cost of crystalloid being used in the DNC was subtracted from the WB-DNC). The cost savings for the total 234 cases for this period of study was approximately USD 117.00. The peak values of both Trop-T and CK-MB release were lower (*p* = 0.001) in the WB-DNC group compared with the DNC group (Table 3). Patients who received WB-DNC showed better renal protection in terms of lower serum creatinine (*p* = 0.001) and blood urea (*p* = 0.006) when compared with the DNC group (Table 3). Similarly, the rise of SGOT after CPB was minimal (*p* = 0.011) in the former group than in the latter group (Table 3).

## 4. Discussion

The present project, based on the medical records of a monocentric tertiary hospital, reviewed the performance of DNC with whole-blood (WB)-DNC in terms of safety and effectiveness. Despite the suitable outcomes of using DNC in adult congenital cardiac surgery, there are concerns regarding its routine use in adult patients [16]. Therefore, it is important to describe the regular use of DNC for CABG surgery and also to address the advantages of WB-DNC in this frame of the study. Hence, we report our experiences with the use of both Del Nido-based cardioplegia (diluted and whole-blood) with the same surgical protocols in the same hospital.

In our study, the degree of hemodilution in terms of median blood hemoglobin concentration observed peri-operatively was lower in the patients who received WB-DNC. Whole-blood cardioplegia alleviates the issue of hemodilution in cardiac surgery by decreasing the load of crystalloid volume being used for the preparation and delivery of crystalloid cardioplegia solutions [17]. Similarly, while comparing with the DNC solution, various studies demonstrated the improved hemoglobin level and better blood conservation in patients delivered cardioplegia with decreasing amount of crystalloid portion [18,19]. Despite non-significant differences due to the use of a strict blood conservation strategy, we found a few patients who need a blood transfusion in the WB-DNC group compared with the DNC (21 vs. 14; *p* = 0.175), thus, showing better results regarding blood conservation. Furthermore, hemodilution during CPB with a hematocrit level of less than 24% is linked with renal injury after cardiac surgery and translates into the worst post-operative outcomes. Therefore, it is important to minimize the hemodilution during CPB for better organ protection [20]. In the current study, the patients showed an improved renal function who received WB-DNC. This might be due to the degree of hemodilution which was less in WB-DNC and provided better renal protection post-operatively.

After cardiac surgery, the elevation of cardiac Trop-T indicates multifactorial perioperative myocardial injury, and it may be considered as a biomarker describing the cumulative intraoperative poor protection of the myocardium [21]. Most patients with an increased post-operative Trop-T measurement suffer perioperative myocardial infarctions [22]. It is suggested that a post-operative measurement of Trop-T or CK-MB may independently predict a patient’s intermediate (12 months) or long-term (>12 months) risk of death or a major cardiovascular event [23]. In the current study, peak values of Trop-T and CK-MB release from cardiac muscles were significantly lower in the WB-DNC-administered patients. Similarly, others also reported elevated concentrations of CK-MB in crystalloid cardioplegia patients [24,25]. Duan and his co-workers also described lower Trop-T levels in patients to whom myocardial protection was achieved using whole-blood cardioplegia [26]. A recent meta-analysis of 11 studies comprising 5798 participants showed that whole-blood cardioplegia was also linked with a decreased CK-MB release, spontaneous return of heartbeat, and a shorter ICU stay [27]. Ad and his coworkers found that troponin was significantly lower at 12 and 24 h post-cardiac surgery in patients that received classical Del Nido cardioplegia (with the same composition that we used) compared to whole-blood cardioplegia in a randomized clinical trial. However, contrary to our study, Ad and his fellows used intermittent whole-blood cardioplegia [28].

Systemic inflammation-induced generalized vasodilatation is a common response of CPB. Catecholamines have been most commonly used after cardiac operations because the low-output syndrome is common during the first hours after the cardiac intervention. Furthermore, vasopressin agents used to combat this might have a negative impact on native and graft coronary flow [29]. The inotropic support needed after cardiac surgery may be related to morbidity and mortality in adult patients [30]. Therefore, certain benefits of less inotropic support indirectly related to better cardiac activity indicate improvement in graft patency, post-operative myocardial perfusion, the incidence of morbidity and mortality, and lesser ICU stay [31,32]. Less inotropic support was required in patients using whole-blood cardioplegia [33]. In our cohort, the need for inotropic support was less in the WB-DNC patients as we infused a higher amount of both adrenaline and dopamine in the patients who were treated with DNC. It shows better myocardial protection and early improvement in cardiac activity after weaning off CPB by the use of WB-DNC.

Our data expressed that the WB-DNC was associated with a reduction in the cost of cardioplegia relative to the DNC solution that was primarily due to the use of Ringer lactate for preparing the crystalloid cardioplegia solution and also because of lesser in-hospital stay of the patients. Our study is in accordance with Timek and co-workers who found that the use of whole-blood cardioplegia is a cost-effective myocardial protection tool as Del Nido additives can directly be added to blood without the need for a crystalloid solution [34].

Spontaneous return of heartbeat is suggested as a sign of good myocardial protection [35]. In the present study, most of the patients recovered spontaneously in both the DNC (92.00%) and WB-DNC (92.10%) groups. No statistical differences regarding conduction disturbances and the need for a pacemaker were seen between the two groups. The same has also been documented earlier in patients undergoing aortic valve replacement, with or without CABG, and who received either cold crystalloid cardioplegia or cold blood cardioplegia [36]. The comparatively higher rate of spontaneous return of sinus rhythm in our cohort might be due to lidocaine in the basal solution that has been associated to minimize ischemic ventricular fibrillation in cardiac surgery [37]. Onorati and his co-workers proposed the supportive findings that better myocardial protection using whole-blood cardioplegia could be responsible for improved post-operative measurements in diastolic function [38].

We did not find any complications during surgery concerning a cardioplegic arrest or weaning off CPB. No difference in mortality was noted, suggesting that both solutions are safe in this respect. In short, patients of both groups had improved clinical measures of post-operative myocardial recovery despite cooling to 6 °C, including the comparable incidence of post-operative atrial fibrillation, risk of infection, liver injury, and length of ICU stay. In a retrospective study, it was found that five-year survival was 96.2% and 87.4% in patients that received cold blood or crystalloid cardioplegia, respectively [39]. Though our study did not compare the effects of both cardioplegia on the long-term survival of patients, our findings do support the conclusion that the use of WB-DNC versus DNC did not affect mortality (30 days after surgery) and provide better overall clinical results particularly considering minimum post-CPB complications, low morbidity rate, and blood loss. Nevertheless, additional randomized clinical studies are warranted to explore the long-term beneficial effects of using WB-DNC in cardiac CABG.

## 5. Conclusions

Based on our results, the use of WB-DNC in CABG seems to be an appropriate myocardial protection strategy. Due to observational and retrospective design of the present study with patients only from a single center, further studies with a prospective design with multicenter involvement may elicit more conclusive information regarding the efficacy of WB-DNC.

## 6. Limitations

The results of this study need to be interpreted cautiously due to certain limitations. Firstly, this was a single-center cross-sectional observational study conducted retrospectively for a limited time period and the results of this study need to be accepted for a similar set of protocols. Secondly, the clinical outcomes were only recorded during the hospital stay of patients. Therefore, the long-term clinical outcomes could not be considered. Thirdly, strict inclusion criteria might have contributed to bias in the outcomes of the study.

## Figures and Tables

**Figure 1 medicina-57-00918-f001:**
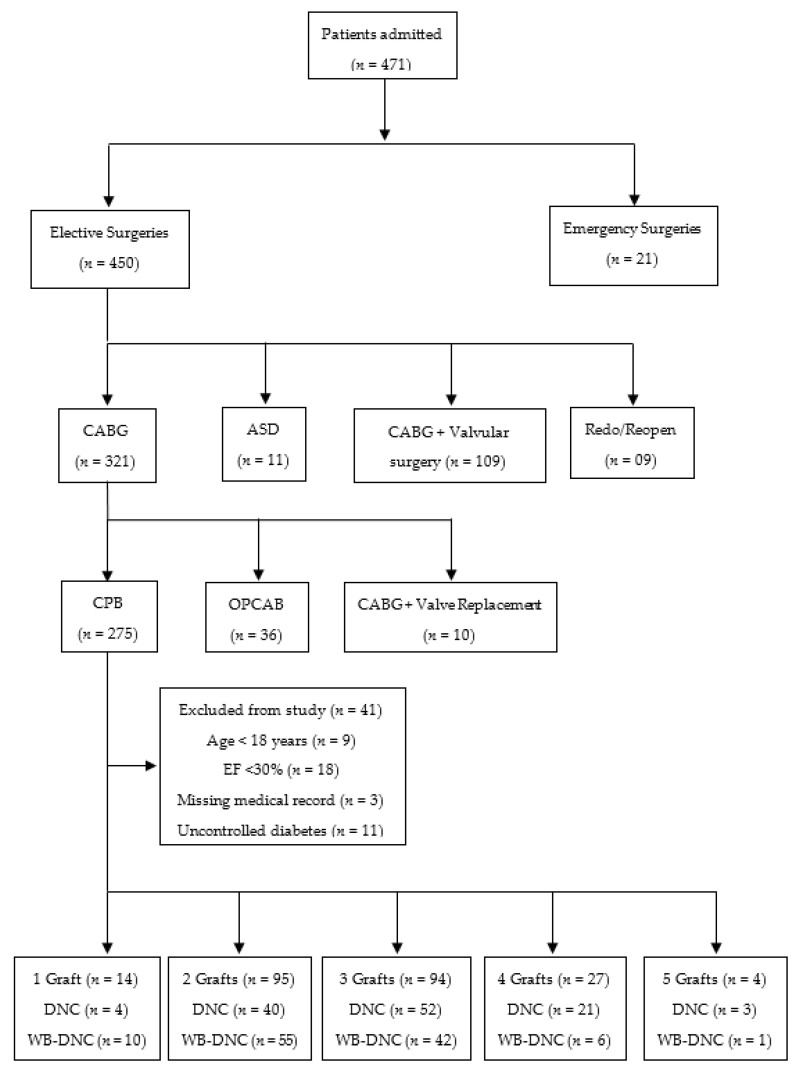
Enrollment of patients. *n* = number of patients; CABG = Coronary artery bypass grafting; ASD = Atrial septal defect; CPB = Cardiopulmonary bypass; OPCAB = Off-pump coronary artery bypass; EF = Ejection fraction.

**Figure 2 medicina-57-00918-f002:**
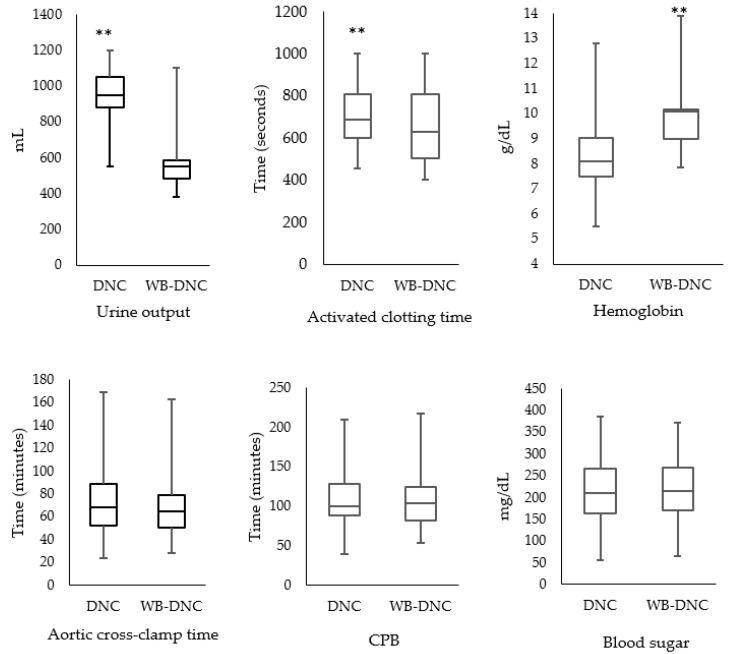
Peri-operative values of urine output, activated clotting time, hemoglobin concentration, aortic cross-clamp time, cardiopulmonary bypass time (CPB), and blood sugar in patients who were either administered Del Nido cardioplegia (DNC) or whole blood (WB)-DNC during CPB. ** represent significant levels at *p* < 0.01.

**Table 1 medicina-57-00918-t001:** Drugs delivered in 1.0 L of Del Nido-based cardioplegia solutions.

Ingredient	Diluted (DNC)	Whole-Blood (WB-DNC)
Base (1.0 L)	(Blood: Ringer Lactate = 1:4)	Blood
KCl (mEq)	26	26
8.4% NaHCO_3_ (mEq)	13	13
25% Mannitol (mL)	13	13
50% MgSO_4_ (mL)	4	4
2% Lidocaine (mL)	6.5	6.5

**Table 2 medicina-57-00918-t002:** Baseline characteristics of patients receiving Del Nido-based cardioplegia.

Parameter	Group; Median (Range)		*p*-Value
	DNC (*n* = 120)	WB-DNC (*n* = 114)	
Age (year)	56 (41–76)	55 (39–72)	0.554
Sex (male) *n* (%)	109 (90.83)	97 (85.08)	0.125
Weight (kg)	72.45 (52.30–126.60)	72.90 (51.50–105.00)	0.666
Height (cm)	165.50 (138–197)	165 (140–193)	0.606
Body mass index (kg/m^2^) *	27.18 ± 5.05	26.74 ± 4.53	0.193
Diabetes mellitus (yes); *n* (%)	74 (61.66)	72 (63.15)	0.460
Smoker (yes) *n* (%)	18 (15)	16 (14.03)	0.491
Ejection fraction (%)	54 (32–67)	54 (30–65)	0.345
EuroSCORE-II	1.50 (1.00–3.00)	1.40 (1.00–3.00)	0.767
Blood urea (mg/dL)	28 (16–55)	32 (18–66)	0.003
Serum creatinine (mg/dL)	1.00 (0.50–1.72)	1.00 (0.50–1.78)	0.930
Serum bilurubin (mg/dL)	0.72 (0.35–2.63)	0.70 (0.29–2.47)	0.931
Hemoglobin (mg/dL) *	14.37 ± 1.93	14.41 ± 1.65	0.117
SGPT (U/L)	25 (14–78)	29 (15–79)	0.081
SGOT (U/L)	30.50 (16–91)	35 (15–87)	0.235
Alkaline phosphatase (U/L)	104 (46–633)	102.50 (49–449)	0.933
White blood cells count (10^3^/μL)	10.20 (4.20–19.50)	10.00 (5.00–19.60)	0.976
Platelets count (10^3^/μL)	226.50 (110–662)	210 (87–665)	0.482

SGPT = serum glutamic pyruvic transaminase; SGOT = serum glutamic oxaloacetic transaminase; * Values are expressed as mean ± SD.

**Table 3 medicina-57-00918-t003:** Postoperative clinical characteristics of patients receiving Del Nido-based cardioplegia.

Parameter	Group; Median (Range)		*p*-Value
	DNC (*n* = 120)	WB-DNC (*n* = 114)	
Blood urea (mg/dL)	38 (20–87)	32.50 (18–97)	0.006
Serum creatinine (mg/dL)	1.24 (0.80–1.90)	1.10 (0.40–1.80)	0.001
SGPT (U/L)	35.50 (17–122)	29 (13–107)	0.065
SGOT (U/L)	73 (20–182)	52.50 (23–183)	0.011
White blood cells count (10^3^/uL)	19.15 (11.70–38.80)	18 (8.90–39)	0.027
Platelets count (10^3^/uL)	160.50 (68–577)	162 (76–521)	0.577
Serum potassium (mEq/L)	4.30 (2.39–5.25)	4.30 (3.29–5.42)	0.868
Peak Troponin-T (ng/L)	394.50 (151–532)	219.50 (150–518)	0.001
Peak creatine kinase myocardial band (μg/L)	25 (15–37)	14 (9–32)	0.001
Atrial defibrillation; *n* (%) yes	12 (7.75)	09 (7.56)	0.370
Spontaneous recovery of heart beat; *n* (%)	108 (90.00)	105 (92.10)	0.573
Pacemaker; *n* (%)	3 (2.50)	0 (0.00)	0.133
Amount of blood loss (mL)	515 (40–1680)	490 (100–1850)	0.681
Blood transfusions; *n* (%) yes	21 (17.50)	14 (12.28)	0.175
Hemoglobin (mg/dL) after weaning of CPB	9.10 (6.40–12.10)	10.10 (6.90–14.15)	0.001
Hemoglobin after shifting to ICU (mg/dL)	10.30 (7.15–14.50)	11.10 (8.00–15.70)	0.001
Activated clotting time (seconds)	107 (90–149)	112 (92–160)	0.156
Adrenaline infusion rate (mcg/kg/min)	0.08 (0.02–0.16)	0.04 (0–0.12)	<0.001
Dopamin infusion rate (mcg/kg/min)	4 (3–10)	4 (2–8)	0.174
Time to extubation (minutes)	120 (52–1935)	120 (24–940)	0.757
ICU stay (days)	5 (1–9)	5 (2–9)	0.701
In-hospital stay (days)	8 (5–14)	7 (5–10)	0.001
Mortality; *n* (%)	2 (1.66)	1 (0.87)	0.519

SGPT = serum glutamic pyruvic transaminase; SGOT = serum glutamic oxaloacetic transaminase.

## Data Availability

The data presented in this study are available on request from the corresponding author (habibrehman@uvas.edu.pk) and are not publicly available due to the identity of patients.

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
