# Peer review of "Efficacy of Whole-Blood Del Nido Cardioplegia Compared with Diluted Del Nido Cardioplegia in Coronary Artery Bypass Grafting: A Retrospective Monocentric Analysis of Pakistan"

_medicina, 2021, doi:10.3390/medicina57090918_

Round 1

Reviewer 1 Report

I would like to thank the editors of Medicina for the opportunity to review the manuscript “Efficacy of Del Nido based whole-blood cardioplegia compared with Del Nido crystalloid cardioplegia in coronary artery bypass grafting: A retrospective monocentric analysis of Pakistan“ by Haider et al.

This retrospective study compares patients who underwent CABG with De-Nido whole-blood (n=120) versus crystalloid (n=114) cardioplegic solutions. The authors conclude that whole-blood Del-Nido solution provided improved myocardial protection and clinical outcome.

The presentation of a whole-blood Del-Nido solution is cost-effective and, therefore, an interesting concept for efficient health care systems. The whole-blood formulation further improves cost-effectiveness. Since prospective studies are scarce (Ad et al, JTCVS), the data from Pakistan are of relevance for the scientific community. Nevertheless, the study has some major methodological flaws which should be revised before consideration for publication.

Abstract:

The abstract is too detailed and should be more concise. Why did the authors use 1:4 (blood-crystalloid) solution?

Introduction

Line 67: Please add references for all cardioplegic solutions

The Introduction is too long, the authors should focus on the used solution with a short explanation why whole blood 4:1 (blood – crystalloid) was used and what the differences to conventional Del-Nido are. In addition, the presented Del-Nido formulation (1:4 blood-crystalloid) should not be called “crystalloid”, since it is blood-mixed. The Reviewer recommends to use simple abbreviations for the two groups (e.g. DNC and WB-DNC) in the whole manuscript. When defined, these abbrev. should be used through the whole manuscript. Do not use different names for the same thing: crystalloid, dilution-based…

Methods

Please add further details about the “Ethical Committee of the Cardiac Surgery Department”. Which Department, which hospital?

Please report the statistics as accepted by your ethics committee.

If the authors describe the “majority of cardiac surgeries were performed by the same...team”, statistics should be provided, otherwise this line is irrelevant.

Was the study registered at clinicalttrials.gov? Please cite.

When was perioperative or postoperative ACT measured?

Statistical Analysis: The authors report multiple primary endpoints, therefore, correction for multiple testing must have been performed. Please further define statistics according to the ethics approval.

Why do the authors perform CPB in moderate hypothermia? Are there technical reasons?

Table 1: Please add the total volume to the table, do not use “crystalloid” for a blood-mixed cardioplegic solution.

Why does the whole-blood solution contain less KCl?

The test-kits/manufacturer of the used Troponin and CK-MB tests should be added.

When were the postoperative outcomes measured? POD1? POD2?

The definition of Mortality is not clear. In-Hospital or 30d mortality?

Study Endpoints: Did the authors correct for multiple testing?

Results

A flow chart of the included patients should be presented.

How were the prices for the solutions calculated?

First, the results should show the primary outcome and then the secondary outcomes.

The reviewer believes that “CKMB” = “… muscle-brain” instead of “myocardial-band”

Inotropic support should not be compared by “duration” but either “yes/no” or “total mcg”.

Time “TO” instead of “OF” extubation.

Further details would be helpful in interpretation of the data: pre- and postop ejection fraction, elective/urgent/emergent surgery, number of anastomoses, type of CAD (1-vessel disease, 2-VD, 3VD…), how many patients needed a perm. Pacemaker?

Discussion

Please include existing prosp. literature on Del Nido whole blood (e.g. Ad et al., JTCVS) into the discussion.

The conclusion needs to be softened. The data are interesting but most of them are only hypotheses generating.

Author Response

Dear Sir

I am greatly thankful for your valuable suggestions and comments.

regards

Reviewer 2 Report

In the manuscript “ Efficacy of Del Nido based whole-blood cardioplegia compared with Del Nido crystalloid cardioplegia in 3 coronary artery bypass grafting: A retrospective monocentric analysis of Pakistan”, Haider et al provide evidence that the whole-blood cardioplegia solution conferred better myocardial protection, improved early clinical outcomes, and also proved to be economical for patients undergoing elective CABG compared with crystalloid cardioplegia solution. The author should provide more detail data to support their conclusion.

The authors just provide data range and p value in all tables. The authors should also provide standard division for each of the data.

In table 3, “Peri-operative selective parameters of patients receiving Del Nido based cardioplegia.” p-value of Urine output (mL) and Hemoglobin (mg/dL) should not be 0.000 in statistic analysis. It should be p < 0.001.

The author should provide scatter plot graph for table 3 and table 4 so that the readers can easily get the information of the data distribution the author provided.

Author Response

Dear Sir

I am very much thankful for your valuable suggestions to improve the quality of the manuscript

Round 2

Reviewer 1 Report

I thank the editors of Medicina for the opportunity to review the revised version of this manuscript.

The authors have made good efforts to improve the manuscript, which is a lot more concise and but not yet suitable for publication. The following comments have to be addressed:

  1. Although literature is usually not very concise in this matter, please differ between “cardioplegic solution” = e.g. Del-Nido and “cardioplegia” = the arrested heart/state/effect of the administration of cardioplegic solution.
  2. Line 79: There seems to be a typo: Was COVID-19 Outbreak in 2020 or 2021?
  3. Statistics: Again, since the authors use multiple primary endpoints, correction for multiple testing has to be performed.
  4. Table 1: Please show only the final molar concentrations of the solutions and delete the comment in the figure legend. This is confusing.
  5. Table 3: total inotropic support: If the authors decide to show total concentrations or yes/no, these should not be pooled but listed separately. Otherwise the reader is not able to distinguish between the different inotropics
  6. Line 195-197: The reviewer does not see the relevance of duration of inotropic support. Does the parameter “total inotropic support” include infusion rates as well as durations?
  7. Figure 2, ACT: It is unclear how this parameter would be sign. different. Which timepoint is shown here?
  8. Figure 2: The figure legend should include the shown timepoint.
  9. Figure 2: use * for p<0.05, ** for p<0.01 etc.
  10. The asterisks for significancies do not match the data in the table and need to be checked again.
  11. For transparency, the reviewer highly recommends to submit the protocol to clinicaltrials.com even for retrospective studies. The registration number should then be mentioned in the manuscript.
  12. Table 3: The authors should provide the timepoint of blood work
  13. Conclusion: The data does not show the differences between 1-5 grafts, therefore, this is not to be included into the conclusion.
  14. Conclusion: The authors should not repeat all results in the conclusion. The reviewer recommends to delete at least “However….setting.” The authors should also describe why clinical trials are warranted (e.g. retrospective study),

Author Response

Dear Reviewer

First of all, the authors are greatly thankful to Reviewer for imparting intellectual input and provided valuable suggestions and comments to improve the quality of the manuscript.

We are extremely thankful to reviewer for providing the learning experience for us. The comments have significantly improved not only the quality of manuscript but also our approach towards the analysis and design of such scientific studies. We greatly appreciate the support and critique of the reviewer regardless of the outcome of this manuscript.

Regards